# Selection of Sweetpotato Parental Genotypes Using Simple Sequence Repeat Markers

**DOI:** 10.3390/plants11141802

**Published:** 2022-07-08

**Authors:** Sonia I. M. Naidoo, Sunette M. Laurie, Assefa B. Amelework, Hussein Shimelis, Mark Laing

**Affiliations:** 1Agricultural Research Council-Vegetable, Industrial and Medicinal Plants (ARC-VIMP), Private Bag X293, Pretoria 0001, South Africa; u12093492@tuks.co.za (S.I.M.N.); slaurie@arc.agric.za (S.M.L.); 2African Centre for Crop Improvement (ACCI), School of Agricultural, Earth and Environmental Sciences, University of KwaZulu-Natal, Private Bag X01, Pietermaritzburg 3209, South Africa; shimelish@ukzn.ac.za (H.S.); laing@ukzn.ac.za (M.L.)

**Keywords:** genetic diversity, gene flow, genotyping, heterotic groups, SSR markers, sweet potato

## Abstract

Knowledge of the genetic diversity and genetic relationship is important in crop improvement. The objective of this study was to determine the genetic diversity of 31 sweetpotato genotypes and furthermore to select distantly related individuals for breeding of superior parental clones. The genotypes (sourced from the Agricultural Research Council, South Africa) originating from Africa and American continent were genotyped using eight highly polymorphic SSR markers. The SSR markers generated a total of 83 putative alleles. The polymorphic information content (PIC) of the tested simple sequence markers varied from 0.73 to 0.91, with a mean of 0.85. At least 11 different alleles were found in 8 loci within the population, with 7 effective alleles per locus. Although high diversity was found among the genotypes, genetic distances among the genotypes were relatively low. Cluster analysis revealed the existence of three distinct genetic groups, and the clustering patterns follow to some extent the geographic origin and pedigree of the genotypes. High gene flow was observed among different sweetpotato accessions. The selected SSR markers were found to be highly polymorphic with high discriminatory power for genetic characterization studies and are useful genomic tool to complement phenotyping of sweetpotato genotypes. Two heterotic groups were found in the study. The heterotic group A was composed of 14 genotypes mainly of South African origin, while the heterotic group B consisted of 17 genotypes of American origin. The two distinct groups were important for the selection of breeding clones that were distantly related to be used as parental clones in the advancement of traits of interest.

## 1. Introduction

Sweetpotato, *Ipomoea batatas* (L.) Lam, is an important root crop cultivated in 117 countries worldwide [1]. It is a staple food in Africa, Asia, the Caribbean, and South America [2,3]. Sweetpotato plays an important role as food and nutrition security. It is a source of income for many communities in the developing countries [4]. It originated in Central America [5,6]. The greatest genetic diversity of sweetpotato, including wild relatives, is in Central America, making it the primary centre of diversity and most likely the centre of origin of the species [7,8,9,10]. Many sweetpotato landraces are found in East Africa, which is considered to be the secondary center of diversity [6,11]. It belongs to the Convolvulaceae family (morning glory vines), which comprises more than 1000 species, of which *I. batatas* is the species of economic interest [2], being a staple crop in many countries of the world [1]. The orange-fleshed sweetpotato, a variant of white-fleshed sweetpatato, is important as source of provitamin A [6].

Sweetpotato is grown as a food crop in most developing countries for its roots and leaves, and has multiple uses in industrial processing [12]. In South Africa, sweetpotato is widely cultivated by resource-poor farmers under low input production systems. It is commonly planted with other crops such as maize or legumes for food security and is sold at local markets [13,14]. The Agricultural Research Council–Vegetable, Industrial and Medicinal Plants (ARC-VIMP) of South Africa is actively involved in the breeding of the sweetpotato to meet the demand for quality and nutritionally balanced improved cultivars.

Through genetic diversity studies, comprehensive information on the magnitude of variation between and within germplasm collection is obtained. This information is fundamental and a precondition for successful breeding and achieving breeding goals [3]. Amongst the approaches to predict the best combinations of parental clones for breeding are the selection of parental lines through genetic diversity analysis measured by genetic distance based on morphological and agronomic characters [15], as well as molecular markers [16]. Although morphological markers are quite inexpensive and easy to use in the genetic diversity studies, they present several limitations due to parallel evolution and a high degree of plasticity, and they are subject to change with diversified environmental conditions and cultivation practices [11,17].

Genetic diversity and phylogenetic analyses using molecular tools are very useful for effective selection and use of germplasm in the planning and execution of breeding programs [17]. Simple sequence repeat (SSR) markers are efficient markers for genetic diversity studies. These markers are widely distributed in eukaryotic genomes, are highly polymorphic, and deliver more information per unit assay than any other marker system. The SSR markers are very suitable for studying variation in a population, even of closely related individuals [17,18]. These markers have been successfully applied in sweetpotato diversity analysis studies [16,17,19,20,21,22].

A comprehensive germplasm collection is preserved at the ARC-VIMP, with more than 360 sweetpotato genotypes from the Americas, Asia, and Africa. However, these genetic resources are yet to be systematically characterised using DNA markers. Few studies on the genetic diversity using molecular markers were performed for most of the ARC sweetpotato genotype collection, and therefore limited information is available on the genetic relatedness of the genotypes. The DNA fingerprinting was performed by Ref. [23] on 25 ARC sweetpotato genotypes and 6 local landraces using five SSR markers, while Ref. [24] used SSR markers for fingerprinting of individual plants of three ARC cultivars. Laurie and coworkers [25] fingerprinted nine orange-fleshed and three cream-fleshed ARC sweetpotato cultivars and elite clones using seven tested SSR markers. Phylogenetic analysis of eight ARC parental clones using SSR markers was performed by Ref. [26] and later Ref. [27] studied the genetic diversity of 30 orange- and cream-fleshed sweetpotato accessions using random amplified polymorphic DNA (RAPD). Therefore, the objectives of this study were to characterize selected sweetpotato genotypes using polymorphic SSR markers and identify distinct and functional groups of sweetpotato clones distantly related to be used in the prediction of progeny performance and selection of elite parents for hybridization.

## 2. Material and Methods

### 2.1. Plant Material

For the scope of the study and due to the limitation of time and resources, various pre-selection approaches were implemented. The plant material used in the present study was selected based on the results from morpho-agronomical characterization of local and international sweetpotato germplasm from South Africa collection at ARC-VIMP in a field trial conducted from January–June 2015 [15]. A pre-requisite for the sample size of the parents being genotyped was the flowering ability of the lines. Following the phenotypic characterization of a larger set of sweetpotato lines [15] and further selection based on the yield, root protein content and flesh colour were done. Only 31 genotypes met the requirements of the important morphological traits in terms of flowering ability, yield potential, flesh colour, and protein content (Table 1). The selected genotypes consisted of 23 breeding clones from South Africa, 4 cultivars from the United States of America (USA), 3 and 1 cultivar/s from Mozambique and Kenya, respectively. The selected genotypes encompassed sweetpotato clones with varying root flesh colour ranging from white to orange and a protein content ranging from low to high.

### 2.2. DNA Extraction and SSR Genotyping

Genomic DNA was extracted from samples of two leaf discs obtained from a young and healthy leaf of three individual plants of each genotype using the modified Cetyl trimethylammonium bromide (CTAB) method, described by [28]. DNA concentration was quantified by nanodrop2000c and diluted to 100 ng μL^−1^.

A set of 10 polymorphic SSR markers (Integrated DNA Technology, Coralville, IA, USA; Table 2), previously used in sweetpotato diversity analysis [11,20,26,29,30] were selected based on their high polymorphic information content (PIC) values.

The microsatellite region was amplified using EconoTaq PLUS 2x Master Mix Catalog No. 30035 (Lucigen, Middleton, WI, USA). A PCR reaction was performed using a total volume of 20 µL consisting of 10 ng of genomic DNA, 10 mM of forward primer and reverse primer, 10 μL of Econotaq PLUS 2x Master Mix (AMRESCO LLC, Cleveland, OH, USA), and nuclease free water. A Gene Amp^®^ PCR system 2700 machine (Applied Biosystems, Carlsbad, CA, USA) was used to amplify individual reactions. Conditions entailed initial denaturation at 95 °C for 5 min, then 45 cycles of denaturation 95 °C for 30 s. Primer annealing was done at optimal temperature for 30 s and primer extension at 72 °C for 1 min. Lastly, a final extension for 20 min at 72 °C was included.

A 1:10 dilution of the fluorescently labelled PCR amplicons, LIZ500 sizing standard and Hi-DiTM Formamide (Catalog No. 4311320, Thermo Fisher Scientific, Carlsbad, CA, USA) was prepared and denatured at 95 °C for 5 min. Following denaturation, the fragments were run on an ABI PRISM™ 3500xl Genetic Analyser (Applied Biosystems, Thermo Fisher Scientific, Carlsbad, CA, USA), 50 cm capillary array, POP-7^TM^. For data analysis, GeneScan™ Software was used, and for data interpretation GeneMapper^®^ v5.0. (Applied Biosystems, Thermo Fisher Scientific, Carlsbad, CA, USA) was used.

### 2.3. Data Analysis

Two approaches were adopted to investigate the genetic structure and diversity among the 31 sweetpotato genotypes. In the first approach, the co-dominant nature of the marker (fragment length polymorphism) was used to determine the genetic parameters such as total number of alleles per locus (N_a_), number of effective alleles per locus (N_e_), allelic richness (A_r_), observed heterozygosity (H_o_), and average gene diversity (H_e_) and inbreeding coefficient (F_IS_) were determined using the protocol of [31] using the software GENALEX version 6.5 [32]. PIC, which is a measure of the usefulness of each marker in clearly distinguishing individuals in a population calculated following [33]:$$PIC_{i}=1-\sum_{j=1}^{n}P_{ij}{}^{2},$$
where $PIC_{i}$ is the *PIC* of a marker *i*; $P_{ij}$ is the frequency of the *j*th pattern for marker *i*, and the summation extends over n patterns.

To examine the population differentiation pairwise F_ST_, genetic distance, identity, and gene flow were calculated for each predetermined population based on flesh colour and protein content. Furthermore, analysis of molecular variance was AMOVA was performed using GENEALEX software. An indirect estimate of the gene flow (Nm) was calculated using the formula:$$\mathrm{Nm}=0.25\;\left(1-F_{\mathrm{ST}}/F_{\mathrm{ST}}\right)$$
where F_ST_ is the F-statistic for genetic differentiation calculated according to Wright’s original derivation [34].

In the second approach, cluster analysis with Ward’s method was performed to established similarity between the entries/samples. Ward’s method is a criterion applied in hierarchical cluster analysis. Ward’s minimum variance method is a special case of the objective function approach originally presented by Ref. [35]. Ward suggested a general agglomerative hierarchical clustering (AHC) procedure, where the criterion for choosing the pair of clusters to merge at each step is based on the optimal value of an objective function. To apply a recursive algorithm under this objective function, the initial distance between individual objects must be (proportional to) squared Euclidean distance. The initial cluster distances in Ward’s minimum variance method are therefore defined to be the squared Euclidean distance between points and calculated using the below formula. The agglomerative hierarchical clustering procedure was performed using XLSTAT (Version 2015.1.03.15485, Addinsoft, Paris).
$$d_{ji}=d(\left\{x_{i}\right\},\{x_{j}\})={\left|\left|x_{i}-x_{j}\right|\right|}^{2}.\;$$

## 3. Results

### 3.1. Genetic Diversity of the Selected Sweetpotato Genotypes

The estimated genetic diversity parameters measured in 31 sweetpotato genotypes using 8 SSR markers are presented in Table 3. Two SSR markers (IB-316 and IB-297) did not amplify any fragments in certain genotypes and were therefore excluded from the data analysis. The allele sizes varied from 81 bp for marker IBSSR18 to 310 bp for marker 690524. Locus IBSSR19 showed the highest variation of allele size that ranged from 195 to 247 bp. The smallest allele size variation was produced by marker IBSSR04 (204 to 226 bp). The major allele frequency per locus varied from 0.11 (IB-248) to 0.34 (IB-242), suggesting an even distribution of alleles among the genotypes.

A total of 83 putative alleles of different sizes were detected. The number of alleles per locus varied from 6 (IB-242) to 15 (IBSSR19), with a mean of 11.9, while the effective number of alleles per locus varied from 3 to 11, with a mean of 7 alleles per locus. The high number of observed and effective alleles implied high levels of polymorphism (high variability of the clones) and good marker choice. A significantly high variation was detected on observed heterozygosity, ranging from 0.59 to 1.00, with a mean of 0.88 among the tested genotypes. The average gene diversity varied from 0.74 to 0.93, with a mean 0.86. The highest gene diversity (He) was scored from marker IB-286 and the smallest by IB-242. A highly significant variation ranged from −0.37 to 0.32, with the mean of −0.04 also observed on fixation index (F_IS_).

In this study, the PIC values varied from 0.73 to 0.91, with a mean of 0.85, suggesting a high discriminatory power of the SSR markers in distinguishing the genotypes. The observed high H_o_ and F_IS_ were expected because sweetpotato is highly heterozygous due to its mode of reproduction and propagation system. In this study, 63% of the markers had negative F_IS_ values. The negative fixation index values indicated the presence of many heterozygotes in the population. For example, for locus IB-242, 74% of the sweetpotato clones were expected to be heterozygous at the specific locus under random mating conditions; however, 100% of the clones at this locus were heterozygotes. All measured genetic parameters suggested that a high level of genetic diversity was present in the tested sweetpotato population.

### 3.2. Within and among Population Variation

Subgrouping of the population was based on the root protein content (RPC) levels (low, intermediate, and high) and root flesh colour (orange, white, cream, and yellow). The genetic diversity within and among the 31 sweetpotato genotypes classified by root protein content and flesh colour is presented in Table 4. High gene diversity was observed among the genotypes from the three groups. A mean of eight alleles was observed per locus. The Shannon information index ranged from 1.48 to 1.93, with a mean of 1.80.

No significant variation was observed between sub-groups with high and intermediate RPC, but highly significant variation was observed between these two sub-groups and the sub-group with low RPC for all the genetic parameters. Clones with high and intermediate RPC levels had significantly higher levels of variation for most of the genetic diversity parameters except H_o_, signifying that there is a high level of genetic diversity maintained in these sub-groups. The mean number of detected alleles (Na) was higher for clones with a high RPC, followed by clones with an intermediate RPC. Clones with a low RPC showed the highest H_o,_ suggesting that this sub-group is highly heterozygous as compared to the two other sub-groups. The highest private allele (17) was detected from the sub-group with high RPC followed by intermediate RPC (14).

Significantly high levels of genetic variation were also detected among the three sub-groups based on flesh colour in all the genetic parameters studied. The number of observed, effective, and private alleles and Shannon’s information index (I) were significantly higher for the sub-group with orange flesh colour, while the white flesh colour genotypes revealed significantly higher values of observed and expected gene diversity.

Clones with orange flesh were genetically diverse in terms of the number of observed and private allele, suggesting that the orange fleshed clones may contain genetically unique and rare alleles. However, the clones with white flesh were the most genetically divers in terms of the number of allele and allele frequency. Additionally, the large number of alleles observed in the orange flesh sub-group might partly be attributed to the large number of samples. Clones with yellow/cream flesh colour had the lowest values for all the genetic parameters. This could either be attributed to the small number of genotypes sampled, or their low level of genetic diversity.

### 3.3. Gene Flow and Genetic Distances

The gene flow (Nm), genetic differentiation (F_ST_), genetic distance (GD), and genetic identity (GI) estimates across sub-groups based on root flesh colour and RPC is presented in Table 5. Gene flow for root flesh colour among the sub-groups varied from 6.57 to 9.57, which were high according to the interpretation guidelines [37,38]. The gene flow from orange fleshed to yellow/cream fleshed sweetpotato was the highest (9.57), followed by gene flow from orange fleshed to white fleshed sweetpotato (8.39). The lowest gene flow was from white fleshed to yellow/cream fleshed sweetpotato genotypes (6.57). Similarly, the gene flow among the sub-groups of RPC varied from 4.72 to 9.14. The gene flow from high to intermediate RPC sub-group was the highest with 9.14, followed by the gene flow from intermediate to low RPC (5.33). The lowest gene flow value was from high RPC sub-group to genotypes with low RPC (4.72).

The average genetic distance among the genotypes was calculated using the method of Ref. [32] and was between 0.00 to 0.05. Genetic distances between white and orange fleshed sweetpotato sub-groups was relatively high at 0.05, while the genetic distance between yellow/cream and orange fleshed was the lowest (0.00). The genetic distance between yellow/cream and white fleshed sub-group was 0.02. In the case of the sub-grouping based on RPC, the average genetic distance varied from 0.05 to 0.14. The highest genetic distance (0.14) was between high and low RPC and the lowest (0.05) was detected between intermediate and low RPC. Genetic distances were higher among the different sub-groups of RPC compared to the sub-groups based on the root flesh colour.

Based on the root flesh colour, the genetic differentiation between the sub-groups varied from 0.03 to 0.04, and based on RPC ranged from 0.03 to 0.05. According to standard guidelines for the interpretation of genetic differentiation, the genetic differentiation in all the sub-groupings was moderate [39]. The relatively narrow genetic distance observed and the low genetic differentiation among the different sub-groups could be attributed to the high gene flow observed among sub-groups.

Analysis of molecular variance (AMOVA) among and within sweetpotato populations is presented in Table 6. The AMOVA among sweetpotato accessions based on flesh colour and root protein content revealed that highly significant variation (98%) was observed within individuals with less contribution from population and among individual variations. Wright’s F-statistic was used to determine the deviation of the Hardy–Weinberg expectation within the population. Highly significant values (*p* < 0.001) were observed for the overall fixation index (F_it_) and were significant (*p* = 0.023) for the inbreeding coefficient (F_is_). No significant variation was observed among the subpopulation due to high gene flow and the use of common breeding stocks.

### 3.4. Genetic Relatedness of Genotypes

Ward’s method, agglomerative hierarchical clustering (AHC) based dendrogram constructed using the Euclidean’s dissimilarity coefficient from binary data of 8 SSR markers, is presented in Figure 1. The scale of dissimilarity is interpreted as 0 = 0% dissimilarity and 35 = 100% dissimilarity. The AHC dendrogram grouped the 31 sweetpotato genotypes into 3 main clusters: 1, 2, and 3 (P1–P3). Euclidean dissimilarity coefficients varied from 5 (14%) to 30 (85%).

P1 consists of two sub-clusters and is comprised of nine genotypes including ‘Bophelo’, ‘2007-17-1’, ‘199062.1 x Ndou’, ‘2010-5-4’, ‘1990-10-2’, ‘2011-10-2’, ‘1988-7-7’, ‘2012-8-4’, and ‘Impilo’. P2 consists of two sub-clusters with a total of five genotypes, namely ‘Mvuvhelo’, ‘2000-12-16’, ‘2005-12-5’, ‘1988-20-1’, and ‘1986-35-1’. P1 and P2 consisted mainly of genotypes from South Africa except for ‘2012-8-4’, a genotype originating from Mozambique. P3 consisted of 17 genotypes that were sub-divided into 3 sub-clusters. The genotypes grouping within Cluster 3 consisted of a mixture of introduced genotypes (‘Resisto’, ‘Hernandez’, ‘Bonita’, ‘Maria Angola’, and ‘1981-27-1204’), South African (‘2008-8-5’, ‘2008-3-1’, ‘Monate x 1999-5-1 (1)’, ‘1987-2-1’, ‘2005-5-5’, ‘1985-7-1’, and ‘1987-19-5’) and African genotypes (‘Melinda’, ‘K135’, ‘2012-29-4’, and ‘2012-18-2’). Most of the South African genotypes in this group are somehow related to the introduced genotypes.

P1 and P2, containing 14 genotypes, were best combined to form heterotic group A (mainly South African genotypes) while P3, consisting of 17 genotypes, was allocated to heterotic group B (mostly introductions from South and North America). Further selection based on the flowering ability were implemented on the 2 heterotic groups and 11 genotypes from each group were identified to be the female and male parental clones for the subsequent implementation of an accelerated breeding scheme, and genetic analysis of quantitative and nutritional traits. The female parents include: ‘1988-7-7’, ‘1988-20-1’, ‘1990-10-2’, ‘2000-12-16’, ‘2005-12-2’, ‘2007-17-1’, ‘2010-5-4’, ‘2012-8-4’, ‘199062.1 x Ndou’, Bophelo, and Impilo. The male parents include: ‘1981-27-1204’, ‘1984-3-66’, ‘1987-2-1’, ‘1985-7-1’, ‘1987-19-5’, ‘2005-5-5’, ‘2008-3-1’, ‘2008-8-5’, ‘2012-29-4’, ‘Monate x 1999-5-1’, and ‘Melinda’.

## 4. Discussion

This study analyzed the genetic diversity and relatedness of selected sweetpotato genotypes using SSR markers. SSRs are abundant in plant genomes, co-dominantly inherited, suitable for automation, and are easily transferable across laboratories, making them very useful for genetic diversity studies in hexaploid species like sweetpotato. The SSR markers used in the present study were highly polymorphic, revealing a mean of 11 polymorphic alleles per locus. Similar results were reported by Refs. [10,40,41,42]. Genetic diversity studies by Ref. [40] assessed 167 Puerto Rican sweetpotato genotypes using 23 SSR markers. Ngailo et al. [41] assessed the genetic diversity of 48 Tanzanian sweetpotato genotypes using 9 polymorphic SSR markers. Furthermore, studies by Ref. [10] also used 8 SSR markers to assess the genetic diversity of 68 sweetpotato genotypes. Similar results were found in the studies by Ref. [10], who detected 89 alleles with an average of 11.12 alleles per locus. The mean number of polymorphic alleles in the present study was higher (11.9) than those reported by Refs. [30,43]. These authors reported an average of 6 and 9 alleles, respectively. Differences in the mean number of polymorphic alleles per locus is likely to be influenced by the number of samples, quality of DNA, number, polymorphism, and discriminatory power of the SSR markers selected and DNA fragment resolution [44].

In this study, the selected SSR markers were discriminatory and informative, as it was revealed by their high PIC values, which ranged from 0.73 to 0.91, the mean being 0.85. The high discriminatory power of SSR markers in hexaploid sweetpotato was demonstrated in the studies by Refs. [20,45], using 4 SSR markers to study the diversity of 57 East African (Tanzanian, Kenyan, and Ugandan) and Tanzanian sweetpotato genotypes, respectively. In the guidelines given by Ref. [46], values provide an indication of the discriminatory power of a marker, and values higher than 0.5 indicate that the markers are highly informative and that heterozygosity is present within a population. This further suggests that PIC is a descriptive measure of genetic diversity. The PIC values found in this study were high for all the loci, indicating a high level of genetic diversity in the studied sweetpotato genotypes. This is in agreement with the mean PIC result obtained by Ref. [47] of 0.85, in their study using 10 SSR markers to assess diversity in 22 sweetpotato genotypes. Studies on the genetic diversity in sweetpotato by Ref. [48] in Burkina Faso and Ref. [40] in Puerto Rico reported mean PIC values of 0.73, which was slightly lower than the results of present study.

In the present study, the selected SSR loci revealed large number of observed allele (mean N_a_ = 11.9) with small major allele frequency (mean M_AF_ = 0.22), which indicated that the alleles were evenly distributed among the tested genotypes, suggesting the presence of wide genetic diversity among sweetpotato. Similarly, the high-observed heterozygosity (mean Ho = 0.88) and the low inbreeding coefficient (mean F_IS_ = −0.04) values confirmed that sweetpotato is a highly heterozygous species. This is likely attributable to its outcrossing and hexaploid nature [49]. Sweetpotato asexual reproduction (vegetative propagation) and self-incompatibility are mechanisms that enable the crop to maintain its high genetic variability and diversity [44,50].

The mean Shannon diversity value observed in this study was 1.83, slightly higher compared to the findings reported by Refs. [16,21], with mean values of 0.45 and 0.43, respectively. The results obtained in the present study were lower than the value of 2.69 reported by Ref. [51]. Genetic distances within genotypes of the sub-groups based on the flesh colour (white, cream/yellow, and orange) and on RPC (low, intermediate, and high) of the population were mostly low in value, the highest value for both sub-groups being of 0.05. Gichuki et al. [8] and Yada et al. [52] reported lower values of genetic diversity 0.18 and 0.57, respectively.

High gene flow values were found among sub-groups based on flesh colour (6.57 to 9.57) as well as RPC (4.72 to 9.14), and were higher than the values of 0.36 to 2.18 reported by Ref. [41]. The narrow genetic distance observed among the different sub-groups of protein content and root flesh colour could partly be attributed to the high gene flow observed in this study. Gene flow is the population parameter that measures the population structure, which indicates the gene migration between groups or amongst populations, causing changes in the allele frequency [53]. The high level of gene flow can be explained by the constant exchange of sweetpotato genetic material between various breeding programs as well as farmers in different regions [11]. The gene flow is also influenced by the level of farmer selection and intra-specific introgression [40], and the pollination mechanism in breeding [54]. High levels of cross-pollination results from the high level of self-incompatibility in sweetpotato, which promotes a high level of gene flow among genotypes. Cross-compatibility promotes the maintenance of high levels of variability and genetic diversity in sweetpotato [41,55].

Cluster analysis revealed the presence of three distinct groups among the studied sweetpotato genotypes, suggesting the existence of a wide range of variation for breeding and strategic conservation [43]. The two clusters (P1 and P2) mainly composed of African genotypes and P3 is composed mainly of American genotypes. Similarly, Ngailo et al. [41] found 3 major genetic clusters among 48 Tanzanian sweetpotato genotypes, although no particular geographical grouping was observed in their study. Diversity analysis studies of sweetpotato genotypes performed by Refs. [20,25,55] likewise found no association between the genotype grouping and the geographic origin. Gichuru et al. [45] studied the diversity of East African genotypes and found that genotypes from Tanzania clustered together, suggesting that these genotypes were morphologically and genetically distinct from the Kenyan and Ugandan genotypes. Using random amplified polymorphic DNA (RAPD) markers, Selaocoe et al. [27] detected clustering of South African germplasm based on country of origin and flesh colour. Likewise, Tumwegamire et al. [19] reported clustering based on the geographic origin, in which the introduced genotypes were clustered separately from East African genotypes. This may be attributed to the difference in the specific breeding objectives such as yield, nutritional value, and tolerance to biotic and abiotic stresses targeted in each region.

In the present study, genotypes from genetic clusters P1 and P2 were linked through geographical origin, both comprising of genotypes of African origin, and were therefore combined to form heterotic Group A. Genetic cluster three (P3) formed by American exotic cultivars as well as their progenies from either direct or polycrosses involving these cultivars, constitute heterotic Group B. As reported by Ref. [3], two distinct groups were found when a set of sweetpotato clones were tested using SSR markers. The two sets of parents were distinguished by the difference in genetic background (origin) and constituted two heterotic groups. Parental clones from the two groups were later used for heterosis exploiting breeding schemes (HEBS) in sweetpotato. Likewise, with the identification of two heterotic groups A and B, selected parental clones were crossed to implement HEBS [56].

Although exploitation of heterosis is well known in other crops such as maize [57], it is becoming a popular concept in sweetpotato breeding as more studies are conducted using SSR markers to determine genetic diversity for the selection of diverse parental lines that are equitably distantly related. Nikiema et al. [58] studied heterosis in sweetpotato progenies from bi-parental crosses between distantly related parental clones. As suggested by Ref. [59], heterosis for yield and other traits is a function of heterozygosity at a large number of loci. Crossing less-related lines or populations generally augments the number of heterozygous loci and increases the level of heterosis observed in crosses, at least over a wide range of genetic diversity [60]. Moll et al. [61] proposed that heterosis may become predictable when molecular markers are used to determine genetic distance and relatedness.

## 5. Conclusions

SSR markers proved to be robust and very informative markers for genetic diversity and relatedness studies on selected sweetpotato genotypes. In this study, a total of 83 putative alleles, with a mean of 11 alleles per loci were observed. Cluster analysis revealed the existence of two heterotic groups. The identified heterotic groups A and B will allow for further selection of parental clones that are genetically diverse and distantly related, which will enable the breeders to exploit heterosis or hybrid vigour. The generated F_1_ progenies will be used as a study population to which an accelerated breeding scheme and genetic analysis of important nutrient traits and yield components will be applied. SSR markers proved to be robust and informative markers capable of determining the genetic diversity and relatedness of sweetpotato genotypes and sub-populations. SSR markers were used successfully and effectively to discriminate heterotic groups within the sweetpotato germplasm maintained in South Africa. This information can be used in future sweetpotato breeding and conservation activities and to advance the sweetpotato breeding program at ARC-VIMP of South Africa. However, a further study is required to confirm the heterotic grouping with a large number of sweetpotato accessions.

## Figures and Tables

**Figure 1 plants-11-01802-f001:**
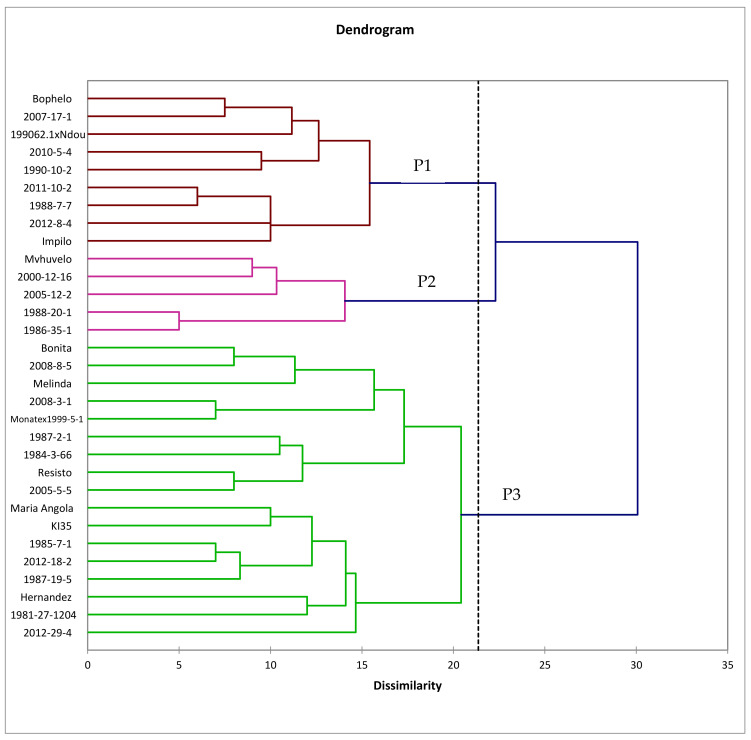
Agglomerative (Ward’s method) hierarchical clustering using Euclidean dissimilarity coefficient on 31 selected sweetpotato genotypes. P1, P2, to P3 denote Cluster 1, Cluster 2, and Cluster 3, respectively.

**Table 1 plants-11-01802-t001:** Sweetpotato genotypes used for SSR genetic diversity analysis and a summary of their pedigree, origin, and agronomic traits.

Accession	Pedigree	Origin	^1^ Flowering Rate	^2^ RPC	^3^ RFC	^4^ RY	^5^ RDMC
Monate x 1999-5-1(1)	1989-17-1 x 1999-5-1 (ex Excel/USA)	RSA x USA	Int	High	Pale-orange	Int-High	High
1981-27-1204	PN USA	USA	Int	Int-High	Orange	Low	Very low
1984-3-66	1981-27-474	RSA	Abundant	Int	Yellow	Int-Low	Int
1985-7-1	Blesbok	RSA	Abundant	Int	White	Int-High	Int-Low
1986-35-1	1981-27-537	RSA	Abundant	Int	White	Int-High	Int-Low
1987-19-5	Bosbok	RSA	Abundant	Int	Cream	Int	Int
1987-2-1	1984-2-201	RSA	Abundant	Int-High	Yellow	Int	Very low
1988-20-1	1985-7-1	RSA	Abundant	Int-High	Cream	Very high	Very low
1988-7-7	Bosbok	RSA	Abundant	Very low	White	Int-Low	Int
1990-10-2	1985-10-5	RSA	Abundant	Very low	White	Int	Int-Low
199062.1 x Ndou	199062.1 x Ndou; 1992-7-2	Peru x RSA	Sparse	Low	Yellow/orange	Int-High	Int
2000-12-16	1992-2-3	RSA	Abundant	Very high	Cream	Int	Int-High
2005-12-2	Khano; 1984-2-201	RSA	Abundant	High	White	High	Very low
2005-5-5	2000-6-3 x Resisto *	RSA	Abundant	Very low	Orange	High	Int
2007-17-1	2001-5-2; 1992-4-1	RSA	Abundant	Int	White	Int-High	High
2008-3-1	1985-6-3 x Rose Centennial *	RSA	Int	Int-Low	Orange	Int-High	Low
2008-8-5	Ndou	RSA	Abundant	Int	Orange	Int	Int
2010-5-4	2004-11-8; 1997-14-18	RSA	Int	High	Orange	High	Int
2011-10-2	2001-5-2; 1992-4-1	RSA	Abundant	Low	Orange	Int	Int-Low
2012-18-2	UW 250A 26-07-06	Moz	Absent/Int	Int	Cream	Int	Int-High
2012-29-4	MUSQ 0674-22	Moz	Int/Absent	High	Cream	Int	Int
2012-8-4	08 ELITE 01-214	USA	Abundant	Int	Orange	Int-High	Low
KI35	-	Kenya	Sparse	Int	Pale orange	Very low	High
Bophelo	1997-14-17	RSA	Absent/Sparse/Abundant	Int	Orange	Int	Int
Hernandez	L70-323	USA	Int	High	Orange	Int-Low	Int
Melinda	W-119	Moz	Sparse/None	Int	Orange	Int-High	Int-Low
Maria Angola	-	Peru	-	High	White	Low	Int
Mvuvhelo	Rose Centennial *	RSA	Absent/Sparse/Abundant	Int	White	Int-High	Int-High
Resisto	W-56	USA	Abundant	High	Orange	Low	High
Bonita	W-152; Excel	USA	Int	Int	White	Int-Low	Int
Impilo	1985-6-3 x Rose Centennial *	RSA	Abundant	High	Orange	Int-Low	Int

Moz = Mozambique; RSA = Republic of South Africa; USA = United States of America. RPC = Root protein content, RY = Root yield; RFC = Root flesh colour; RDMC = Root dry matter content; Int = Intermediate, - = unknown; * = ex USA. ^1,2,3,4,5^ summary of phenotypic and agronomic data obtained from historic characterization files (ARC data base) on ARC germplasm collection.

**Table 2 plants-11-01802-t002:** List of SSR markers with annealing temperature, expected size, and motif used in the genotyping of 31 selected sweetpotato genotypes.

Marker Name	Primer Sequences	Ta ^1^ (°C)	Expected Size	Motif	Reference
IBSSR04	F: CTC CTT TGC CTC CTT TCA TGC	60	160–216	(GA)11	[26,36]
	R: CCT TGC TCC CCA TTT TCT TG				
IBSSR17	F: ACG TGC AGA CTT AGC CAC AC	56	201–245	(AG)_6_N(AG)_17_	[26,36]
	R: AGG AAG CCA GAT GTT CAG ATG				
IBSSR18	F: GAT CTT GAA TTA GCC CAC	58	90–110	(GA)_7_(AG)_5_(GA)_4_	[26,36]
	R: AGA TGG ATG ACC GTA TGC				
IBSSR19	F: GCG AAT CAA GTC TTT TGT CCA C	65	171–195	(CA)_25_	[26,36]
	R: GGG ACT GTC CTT TGG GTA TG				
IB-242	F: GCG GAA CGG ACG AGA AAA	52	95–135	(CT)_3_CA(CT)_11_	[19,29]
	R: ATG GCA GAG TGA AAA TGG AAC A				
IB-248	F: GAG AGG CCA TTG AAG AGG AA	62	164–177	(CT)_9_(CT)_8_	[19,29]
	R: AAG GAC CAC CGT AAA TCC AA				
IB-286	F: AGC CAC TCC AAC AGC ACA TA	50	90–122	(CT)_12_	[19,29]
	R: GGT TTC CCA ATC AGC AAT TC				
IB-297	F: GCA ATT TCA CAC ACA AAC ACG	58	130–196	(CT)_13_	[26,29]
	R: CCC TTC TTC CAC TTT CA				
IB-316	F: CAA ACG CAC AAC GCT GTC	54	150	(CT)_3C_(CT)_8_	[26,29]
	R: CGC GTC CCG CTT ATT TAA C				
690524	F: AAG GAA GGG CTA GTG GAG AAG GTC	57	240–315	(CT)_13_	[20,26]
	R: CAA GGC AAC AAA TAC ACA CAC ACG				

Ta ^1^ (°C) = annealing temperature in degree Celsius.

**Table 3 plants-11-01802-t003:** Genetic parameters describing the diversity among 31 sweetpotato genotypes based on 8 polymorphic SSR markers.

Loci		Genetic Parameters						
	Expected Frag. Size Range	Na	Ne	Ho	He	F_IS_	PIC	M_AF_
IBSSR04	204–226	10	8.25	0.97	0.89	−0.10	0.88	0.15
IBSSR17	210–235	11	5.34	0.77	0.83	0.05	0.82	0.29
IBSSR18	81–113	14	7.45	0.90	0.88	−0.04	0.87	0.21
IBSSR19	195–247	15	7.34	0.59	0.88	0.32	0.86	0.22
IB-242	137–160	6	3.67	1.00	0.74	−0.37	0.73	0.34
IB-248	157–204	14	10.33	1.00	0.92	−0.11	0.90	0.11
IB-286	105–158	13	11.37	1.00	0.93	−0.10	0.91	0.13
690524	283–310	12	5.55	0.77	0.83	0.06	0.82	0.29
Mean	-	11.88	7.41	0.88	0.86	−0.04	0.85	0.22
SE		1.03	0.91	0.05	0.02	0.07	0.02	0.03

Na = total number of alleles per locus; Ne = number of effective alleles per locus; Ho = observed gene diversity within genotypes; He = average gene diversity within genotypes; F_IS_ = inbreeding coefficient; PIC = polymorphic information content; M_AF_ = Major allele frequency per locus; SE = Standard error.

**Table 4 plants-11-01802-t004:** Genetic diversity within and among the 31 sweetpotato genotypes classified by root protein content and flesh colour.

	Genetic Parameter							
RPC								
Population	N	Na	Ne	I	Ho	He	PA	%P
Clones with high RPC	14.00	9.13	6.39	1.93	0.88	0.85	17	100
Clones with intermediate RPC	12.00	8.63	6.65	1.97	0.86	0.87	14	100
Clones with low RPC	5.00	5.00	4.19	1.48	0.91	0.84	5	100
RFC								
Orange	15.00	9.75	6.81	2.01	0.88	0.86	20	100
White	8.00	7.75	5.96	1.86	0.89	0.87	11	100
Yellow and cream	7.00	6.50	5.17	1.71	0.84	0.85	5	100
Overall mean	10.04	8.00	5.98	1.86	0.88	0.86	-	100
SE	0.84	0.55	0.44	0.08	0.04	0.02	-	0

RPC = Root protein content; RFC = Root flesh colour; N = Number of individuals within each population; Na = total number of alleles per locus; Ne = number of effective alleles per locus; I = Shannon’s information index; Ho = observed gene diversity within genotypes; He = average gene diversity within genotypes; PA = number of private alleles; % P = percentage of polymorphic loci; SE = Standard error.

**Table 5 plants-11-01802-t005:** Pair-wise estimates of gene flow (N_m_), genetic differentiation (F_ST_); genetic distance (GD), and genetic identity (GI) according to sub-groups based on root flesh colour (RFC) and levels of root protein content (RPC).

N_m_			
RFC	Orange	White	Yellow and cream
Orange	-	8.39	9.57
White		-	6.57
Yellow and cream			-
RPC	High	Intermediate	Low
High	-	9.14	4.72
Intermediate		-	5.33
Low			-
F_ST_			
RFC	Orange	White	Yellow and cream
Orange	-	0.03	0.03
White		-	0.04
Yellow and cream			-
RPC	High	Intermediate	Low
High	-	0.03	0.05
Intermediate		-	0.05
Low			-
GD			
RFC	Orange	White	Yellow and cream
Orange	-		
White	0.05	-	
Yellow and cream	0.00	0.02	-
RPC	High	Intermediate	Low
High	-		
Intermediate	0.08	-	
Low	0.14	0.05	-
GI			
RFC	Orange	White	Yellow and cream
Orange	-		
White	0.95	-	
Yellow and cream	1.00	0.98	-
RPC	High	Intermediate	Low
High	-		
Intermediate	0.92	-	
Low	0.87	0.95	-

**Table 6 plants-11-01802-t006:** AMOVA among and within groups of sweetpotato genotypes.

Source	df	SS	MS	Est. Var.	Perc. Var	F-Statistics
Among the groups of individuals	2	6.337	3.168	0.000	0%	F_ST_ = 0.782
Among individual	28	99.309	3.547	0.080	2%	F_IS_ = 0.023
Within individual	31	105.000	3.387	3.387	98%	F_IT_ = 0.001
Total	61	210.645		3.467	100%	

df = degrees of freedom; SS = sum of squares; Est. var. = estimated variance, Perc. Var = percentage variance F_ST_ = genetic differentiation, F_IS_ = fixation index or inbreeding coefficient and F_IT_ = Overall fixation index.

## Data Availability

Data is contained within the article.

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
