# Peer review of "Selection of Sweetpotato Parental Genotypes Using Simple Sequence Repeat Markers"

_plants, 2022, doi:10.3390/plants11141802_

Round 1
Reviewer 1 Report
Dear Authors
The Present manuscript entitled "Genetic Diversity of Sweetpotato Genotypes Using Simple Sequence Repeat Markers" demonstrated the genetic diversity of 31 sweetpotato genotypes to select distantly related individuals for breeding of superior parental clones. The findings identified two heterotic groups, group A was composed of 14 genotypes mainly of 23
South African origin while the heterotic group B consisted of 17 genotypes of American origin and suggested the importance of two distinct groups for the selection of breeding clones that were distantly related to be used as parental clones in the advancement of traits of interest.
The study was very well planned and performed, introduction was quite informative. Methods are properly described and results were discussed nicely with concise conclusion. I do not have any further query or suggestions.
Thank you
Author Response
The Present manuscript entitled "Genetic Diversity of Sweetpotato Genotypes Using Simple Sequence Repeat Markers" demonstrated the genetic diversity of 31 sweetpotato genotypes to select distantly related individuals for breeding of superior parental clones. The findings identified two heterotic groups, group A was composed of 14 genotypes mainly of 23 South African origins while the heterotic group B consisted of 17 genotypes of American origin and suggested the importance of two distinct groups for the selection of breeding clones that were distantly related to be used as parental clones in the advancement of traits of interest.
The study was very well planned and performed, the introduction was quite informative. Methods are properly described and results were discussed nicely with a concise conclusion. I do not have any further queries or suggestions.
Thank you for reading the manuscript and for the kind comments.
Reviewer 2 Report
This manuscript presents an approach to studying the genetic diversity of sweetpotato genotypes. The study was conducted to answer a well-defined research question and the authors were able to support their claims using the conclusions of this study. The only part, I found that is poorly presented, is the methodology section. It is strongly recommended that this section should be elaborated to provide details of methods used to generate the outcomes of this study. Please see some related comments below:
Line 302: This study has only one figure how it can be Figure 5?
Figure: The matrix of Euclidean dissimilarity coefficients should be provided as supplementary data which was used to generate the dendrogram. Also, the programming languages and computer requirements along with the version of the software and the values of parameters should be mentioned in the methodology section.
Additionally, a detailed methodology and/or scripts/codes should be provided that can be used to reproduce the results of this study using provided input data.
Line 396: This is not the right way to add a reference. Please check the entire manuscript for such errors.
In the discussion section, please mention how the results of this study can be employed in future studies to explore this topic further. Also, please mention some relevant open research questions that were not addressed during this study but could contribute significantly to exploring the genetic diversity of crops.
Author Response
This manuscript presents an approach to studying the genetic diversity of sweetpotato genotypes. The study was conducted to answer a well-defined research question and the authors were able to support their claims using the conclusions of this study. The only part, I found that is poorly presented, is the methodology section. It is strongly recommended that this section should be elaborated to provide details of methods used to generate the outcomes of this study. Please see some related comments below:
Thank you for your constructive observations. We have addressed your comments as follows:
Line 302: This study has only one figure how it can be Figure 5?
Response 1: It is a typographic error and is corrected in Figure 1.
Figure: The matrix of Euclidean dissimilarity coefficients should be provided as supplementary data which was used to generate the dendrogram. Also, the programming languages and computer requirements along with the version of the software and the values of parameters should be mentioned in the methodology section.
Response 2: The information the reviewer asked to include is included in lines 140-151.
Additionally, a detailed methodology and/or scripts/codes should be provided that can be used to reproduce the results of this study using provided input data.
Response 3: The detailed information on the methods employed, statistical software used, and the formulas are indicated in the MM on the data analysis sub-section.
Line 396: This is not the right way to add a reference. Please check the entire manuscript for such errors.
Response 4: The entire manuscript has been checked and the mistakes are corrected.
In the discussion section, please mention how the results of this study can be employed in future studies to explore this topic further. Also, please mention some relevant open research questions that were not addressed during this study but could contribute significantly to exploring the genetic diversity of crops.
Response 5: The required information is included in the manuscript on lines 425-428
Reviewer 3 Report
The report lacks novelty and is based on very limited data. Why only 31 genotypes out of more than 360 sweet potato genotypes of the germplasm collection were included in this study? Why only 13 (but only 10 in the manuscript) SSR markers were tested? A total of 83 putative alleles are too few to reliably assess the genetic variability for polyploid sweet potato plants. Why the flesh colour and root protein content data were included?
The paper is not suitable for Plants-MDPI.
Author Response
The report lacks novelty and is based on very limited data. Why only 31 genotypes out of more than 360 sweet potato genotypes of the germplasm collection were included in this study? Why only 13 (but only 10 in the manuscript) SSR markers were tested? A total of 83 putative alleles are too few to reliably assess the genetic variability for polyploid sweet potato plants. Why the flesh color and root protein content data were included?
The paper is not suitable for Plants-MDPI.
Thank you for your thoughtful observations. We have addressed your comments as follows:
Response:
We would like to establish that the main objective of the study was to select parental lines from distantly related genetic groups for breeding superior varieties. This was not general or common diversity studies of the entire ARC germplasm collection. For the scope of the study and due to the limitation of time and resources, various pre-selection approaches were implemented before genetic diversity analysis was performed on the selected potential sweetpotato parents. A pre-requisite for the sample size of the parents being genotyped was the flowering ability of the lines. Following the morpho-agronomic characterization of a larger set of sweetpotato lines [15] and further selection based on the yield, root protein content, and flesh color, 31 genotypes were identified. Although we have 360 sweetpotato genotypes in the entire ARC germplasm collection, the pre-selected 31 genotypes met the requirements of the important morphological traits in terms of flowering ability, yield potential, flesh color, and protein content.
Flowering ability was one of the predetermining traits for the selection of a particular genotype since our objective was selecting lines for breeding, therefore flowering ability and synchronization amongst the sweetpotato lines was very imperative and an important selection factor. Once flowering ability was established, we further selected lines that had high, intermediate, and low protein content, yield, and dry matter content to be able to measure the genetic gain. The information regarding the dissimilarity/relatedness of the parental lines was important to make a final selection of parents that were distantly related to be able to exploit heterosis (hybrid vigor) by crossing two heterotic groups. That was the reason why only 31 genotypes were used for this study. However, we would like to suggest that future studies gather enough funding and resources to be able to perform an exhaustive diversity study where the degree of relatedness among the 360 genotypes will be established for future sweetpotato breeding. In addition, there are many research reports done on many crops only using a few polymorphic SSR loci. Only a few SSR loci can give good discrimination between closely related individuals. SSR markers have high discriminatory power, are quite abundant and multi-allelic, and are also widely dispersed across the genome.
Round 2
Reviewer 2 Report
I have no further major comments. The authors incorporated the previous comments carefully.
Author Response
The authors thank the reviewer for constructive comments
Reviewer 3 Report
Sorry, the revised manuscript still lacks novelty and contains very limited data. It is not suitable for Plants-MDPI. The authors may consider journals related to crop breeding.
Author Response
The authors acknowledge the comment.
We feel the paper does add value to the field of heterosis and speed breeding in sweet potatoes for which there are very few published articles.
